# Effect of chronic kidney disease on outcomes following proximal humerus fragility fracture surgery in diabetic patients: A nationwide population-based cohort study

**Chien-Tien Chen[1], Su-Ju Lin[2], Liang-Tseng Kuo[3,4]\*, Tien-Hsing Chen[3,5], Wei-Hsiu Hsu[3,4], Chi-Lung Chen[4], Pei-An Yu[4], Kuo-Ti Peng[1,3], Yao-Hung Tsai[3,4]**

1 Department of Orthopedic Surgery, Chang Gung Memorial Hospital, Chiayi, Taiwan, 2 Division of Nephrology, Department of Medicine, Chang Gung Memorial Hospital, Chiayi, Taiwan, 3 School of Medicine, Chang Gung University, Taoyuan, Taiwan, 4 Division of Sports Medicine, Department of Orthopedic Surgery, Chang Gung Memorial Hospital, Chiayi, Taiwan, 5 Division of Cardiology, Department of Medicine, Chang Gung Memorial Hospital, Keelung, Taiwan

\* light71829@gmail.com

## Abstract

### Background

The proximal humerus fracture (PHF) is the third most common fragility fracture. Diabetes mellitus (DM) and chronic kidney disease (CKD) are both risks for fragility fractures; however, the interplay of DM and CKD makes treatment outcomes unpredictable. This study aimed to investigate and compare early and late outcomes following proximal humerus fracture fixation surgery in diabetic patients with different renal function conditions.

### Methods

DM patients receiving PHF fixation surgery during 1998–2013 were recruited from Taiwan's National Health Insurance Research Database. According to their renal function, patients were divided into three study groups: non-chronic kidney disease (CKD), non-dialysis CKD, and dialysis. Outcomes of interest were early and late perioperative outcomes. Early outcomes included in-hospital newly-onset morbidities. Late outcomes included infection, revision, readmission, and all-cause mortality.

### Results

This study included a total of 10,850 diabetic patients: 2152 had CKD (non-dialysis CKD group), 196 underwent permanent dialysis (dialysis group), and the remaining 8502 did not have CKD (non-CKD group). During a mean follow-up of 5.56 years, the dialysis group showed the highest risk of overall infection, all-cause revision, readmission, and mortality compared to the non-dialysis CKD group and non-CKD group. Furthermore, subgroup analysis showed that CKD patients had a higher risk of surgical infection following PHF surgery than non-CKD patients in cases with a traffic accident or fewer comorbidities (Charlson Comorbidity Index, CCI <3) (*P* for interaction: 0.086 and 0.096, respectively). Also, CKD

provided by the Taiwan Central Bureau of National Health Insurance, the Department of Health, and managed by the Taiwan National Health Research Institutes. Researchers interested in accessing this dataset can submit a formal application to request access (Taiwan Ministry of Health and Welfare, No. 488, Section 6, Zhongxiao E Rd, Nangang District, Taipei City 115, Taiwan; Email: nhird@nhri.org.tw.

**Funding:** The author(s) received no specific funding for this work.

**Competing interests:** The authors have declared that no competing interests exist.

patients had an even higher mortality risk after PHF surgery than non-CKD patients, in females, those living in higher urbanization areas, or with more comorbidities (CCI $\geq$3) ($P$ for interaction: 0.011, 0.057, and 0.069, respectively).

## Conclusion

CKD was associated with elevated risks for infection, revision, readmission, and mortality after PHF fixation surgery in diabetic patients. These findings should be taken into consideration when caring for diabetic patients.

## Introduction

A proximal humerus fracture is common among the aging population, especially in patients with osteoporosis. It accounts for about 5% of all adult fractures and is the third most common osteoporotic fracture following hip and distal radius fracture [1]. While most proximal humerus fractures are minimally displaced and treated conservatively, more and more displaced fractures are treated surgically [2]. Fracture pattern, quality of the bone, and patient-related factors all affect surgical outcomes after proximal humerus fracture [3]. Despite the recent development of the locking plate technique, complications still frequently occur, including screw penetration, varus collapse, avascular necrosis, nonunion, and deep infection, which may need subsequent surgical treatment [4].

Patients with chronic kidney disease (CKD) are more likely to fall and suffer a fracture. Low bone quality and fracture risk significantly increased in a graded manner with renal function deterioration [5–7]. The 3-year cumulative fracture incidence in women over 65 years of age across the five eGFR groups increased incrementally from 4.3% to 9.6% [7]. A similar tendency was shown in a recent meta-analysis [8], in which the risk of fractures was the highest in patients with stage 5 CKD or dialysis. Also, poor renal function worsened the perioperative outcomes after a fracture. Patients under regular dialysis may have a 3.7-fold higher death rate and 4-fold hospitalization rate than the general population following a fracture [9]. Our previous studies also demonstrated that CKD was associated with poor outcomes following hip fracture and a higher risk of revision, readmission, and mortality [10, 11].

Since diabetes mellitus (DM) is one of the leading causes of renal dysfunction [12], it usually coexists in the same patient with CKD. In addition, DM increases the risk of osteoporotic fracture [13–15] and contributes to poor outcomes after fracture surgery [16–18]. However, the above studies have mainly focused on hip fracture, and information on proximal humerus fracture is scant. Furthermore, the coexistence of CKD and DM may make surgical outcomes unpredictable. Therefore, we designed this study to elucidate this issue. The study aimed to compare perioperative outcomes for proximal humerus fracture in diabetic patients with different renal function degrees. We hypothesized that DM patients with CKD would have less favorable outcomes after proximal humerus fracture fixation surgery than patients without CKD.

## Materials and methods

### Data source

We conducted a nationwide retrospective cohort study using data from the Taiwan National Health Insurance Research Database (NHIRD), covering more than 23 million Taiwanese

(>98% of the population). The NHIRD records gender, date of birth, residential area, income, surgical procedure, disease diagnostic codes according to the International Classification of Disease, Ninth Revision, Clinical Modification (ICD-9-CM) medications. Further information regarding the NHI program and the NHIRD has been reported in previous publications [19, 20]. The study was performed with the Ethics Institutional Review Board of Chang Gung Memorial Hospital (IRB CGMH 103-5040B), whereby the informed consent requirement was waived since the data was anonymous.

## Patient identification

All type 2 DM patients admitted due to proximal humerus fracture were included from January 1, 1998, to December 31, 2013. Diagnoses of DM in NHIRD have been validated based on ICD-9-CM diagnostic codes, with at least four outpatient visits corresponding to an accuracy of 95.7% [21]. Another study has demonstrated that any oral hypoglycemic agents' prescription corresponded to an accuracy of 99% [22]. In that study, the type 2 DM diagnosis was supported by medical records from at least four outpatient visits and any use of glycemic-lowering drugs [21]. The first admission for the proximal humerus fracture was assigned as the index admission if the patient suffered multiple episodes during the study period.

We excluded those with multiple trauma and previous implant-related infections to ensure that patients presenting with first-ever proximal humerus fracture. Patients with other conditions that could jeopardize the postoperative outcomes were also excluded, such as malignancy, history of renal transplantation, and immune deficiency. The included patients were then classified into three groups according to renal functional status: patients with normal renal function (Non-CKD group), patients with impaired renal function but free of dialysis (Non-dialysis CKD group), and patients with current dialysis (Dialysis group). The diagnosis of chronic kidney disease (CKD) was validated with the ICD-9-CM diagnostic code. Current practice in Taiwan mostly follows the National Kidney Foundation K/DOQI clinical guidelines. CKD was defined as either GFR <60ml/min/1.73m$^2$ or kidney damage, either pathologically or with abnormal markers, such as urine or blood [23]. Dialysis status meant that patients qualified for a catastrophic illness certificate (CIC), also recorded in the NHIRD. Ultimately, 10,850 patients remained for inclusion in this study. The flowchart delineating patient inclusion is shown in Fig 1.

## Covariates

The covariates were age, gender, the reason for injury at the time of the index admission, hospital level of the index admission, hospital volume of proximal humerus fractures between 1998 and 2013, monthly income, urbanization level of the patient's residence, surgical duration, comorbidities, and Charlson's Comorbidity Index (CCI). Comorbidities were identified with at least one inpatient diagnosis or two consecutive outpatient records in the previous year [24]. Most comorbidity diagnoses based on ICD-9-CM codes have been previously validated (S1 Table) [22, 24]. Involvement in a traffic accident was recorded in the inpatient claims data. The patient's demographics (age, gender, monthly income, and urbanization level) were recorded in the Registry for Beneficiaries in the NHIRD. Information on surgical duration was captured using the Taiwan NHI reimbursement codes from the inpatient claims data.

## Outcomes

Outcomes of interest included early in-hospital outcomes and late outcomes after discharge. Early in-hospital outcomes, including new-onset venous thromboembolism (VTE), delirium, urinary tract infection (UTI), and pneumonia, were identified using the ICD-9-CM codes of

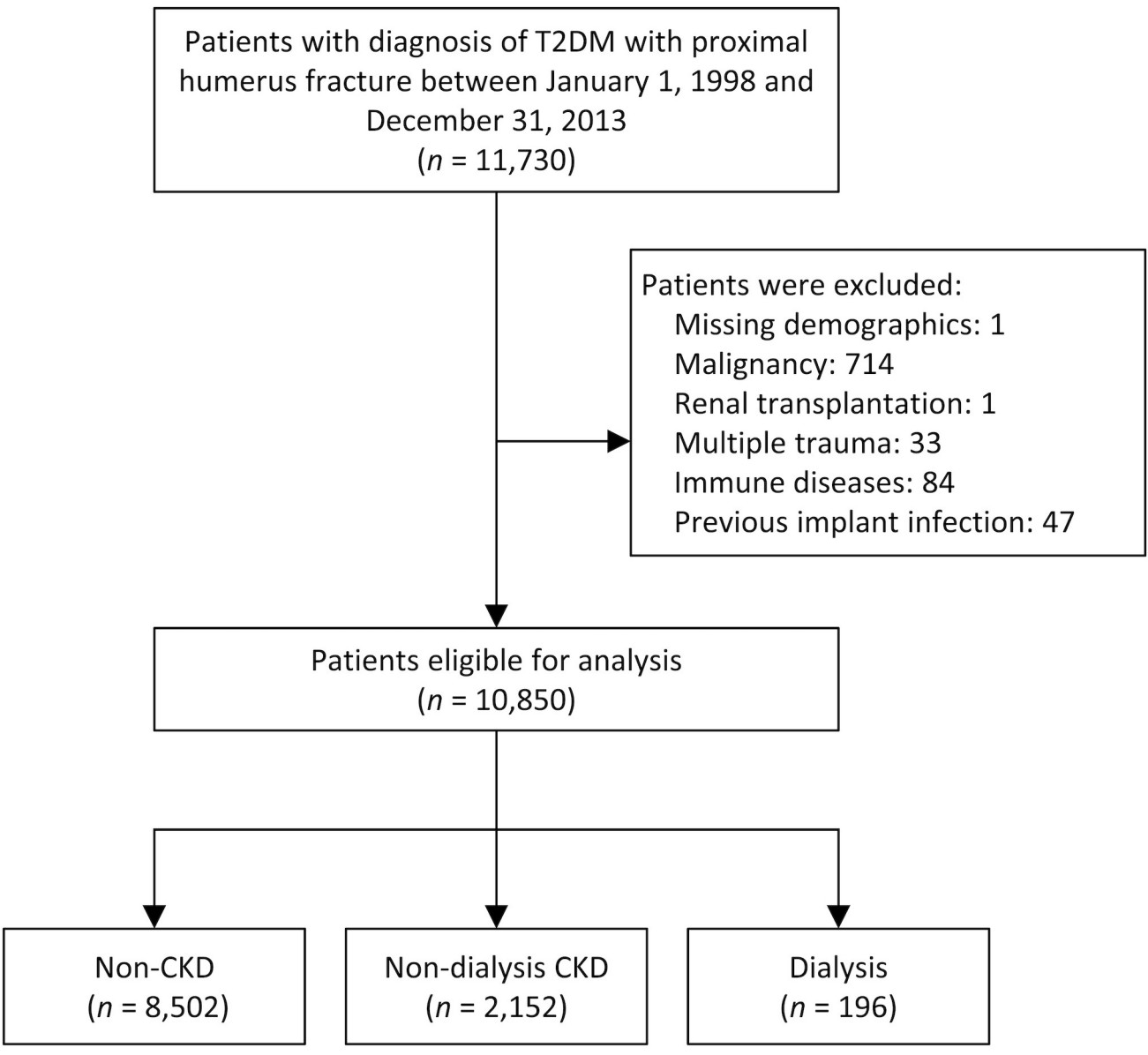

**Fig 1. The flowchart for the inclusion and exclusion of patients.** T2DM = type II diabetes mellitus.

the discharge diagnoses. Debridement, infection, blood transfusion, and intensive care unit stay were detected using the Taiwan NHI reimbursement codes from the inpatient claims data. Information about the length of hospital stay and the medical cost was recorded in the inpatient claims data.

Late outcomes included postoperative infection and PHF revision, as identified by the inpatient NHI reimbursement codes. Postoperative infections included superficial wound infections and deep infections. Superficial infection was defined as readmission for antibiotics administration only, while deep infection required surgical debridement with or without implant removal [11]. PHF revision was classified into infectious or non-infectious, based on whether any postoperative infection occurred three months before revision. Late outcomes also included all-cause readmission and all-cause mortality within 30-days, 90-days, or 1-year,

and at the end of follow-up. Death was identified by withdrawal from the NHI program [25]. Patients were followed from discharge after index admission until the date of events' occurrence, death, or the end of the database (December 31, 2013), whichever occurred first.

## Statistical analysis

Baseline characteristics of patients among these three groups were compared by chi-square test for categorical variables and one-way analysis of variance for continuous variables. Bonferroni multiple comparisons were made when the overall test result was statistically significant. The risk of all-cause mortality during follow-up among the three groups was compared using Cox proportional hazard model. The cumulative risk of non-fatal time-to-event outcomes (infection, revision, or readmission) among the three groups was compared with the Fine and Gray sub-distribution hazard model, in which death during follow-up was considered a competing risk [26]. The potential confounders listed in Table 1 have been adjusted in both Cox and sub-distribution hazard models, except where follow-up duration was replaced with the index date.

Finally, post-hoc subgroup analyses on the outcomes of interest with statistical significance (including overall infection, all-cause revision, all-cause readmission, and all-cause mortality at the end of follow up) were conducted to assess whether the effect of renal functional status on outcomes was consistent across different levels of subgroup variables. Selected subgroup variables were age (categorized into 20–64 years, 65–74 year, and ≥75 years), gender, the reason for injury at index admission, urbanization level (low/moderate vs. high/very high), hospital volume (low vs. high), and CCI score (dichotomized by 3 points). Due to the small sample size of the dialysis group, the non-dialysis CKD and dialysis groups were combined as a CKD group in the subgroup analysis.

A two-tailed $P$ value of $< 0.05$ was considered statistically significant, and no adjustment for multiple testing (multiplicity) was made. All statistical analyses were performed using SAS version 9.4 (SAS Institute, Cary, NC). The adjusted cumulative incidence function and adjusted survival rate were generated using the "%dacif" and "%adjsurv" SAS macros, respectively [27, 28].

## Results

### Baseline characteristics

A total of 10,850 type II DM patients, who underwent proximal humerus fracture fixation surgery, were eligible for analysis between 1998 and 2013. These patients were divided into non-CKD (8,502 patients), non-dialysis CKD (2,152 patients), and dialysis (196 patients) groups (Fig 1).

The mean age was 63–69 years for the three groups, with the non-dialysis CKD group being the oldest. The non-CKD group was most likely to have experienced traffic accidents, while the non-dialysis CKD and dialysis groups followed in sequence. Hypertension was the most prevalent comorbidity for each of the three groups. Compared to the non-CKD group, the non-dialysis CKD group had a higher prevalence of all comorbidities, while the dialysis group had the highest prevalence of heart failure, coronary heart disease, and myocardial infarction. The dialysis group had the highest mean CCI score among the three groups (non-CKD: 2.0 ± 2.0, non-dialysis CKD: 3.0 ± 2.0, dialysis: 6.0 ± 2.0). The mean follow-up duration for the non-CKD, non-dialysis CKD, and dialysis groups was 6.0, 4.1, and 2.7 years, respectively (Table 1).

**Table 1. Baseline characteristics of diabetic patients with proximal humerus fracture according to renal function status.**

| Variable | Non-CKD (n = 8,502) | Non-dialysis CKD (n = 2,152) | Dialysis (n = 196) | P value |
|---|---|---|---|---|
| Age (years) | 63.8 ± 12.8 | 68.5 ± 11.5 [a] | 64.6 ± 10.6 [b] | <0.001 |
| Age group | | | | <0.001 |
| < 60 yrs. | 3,139 (36.9) | 455 (21.1)[a] | 56 (28.6)[a,b] | |
| 60–80 yrs. | 4,624 (54.4) | 1,377 (64.0)[a] | 128 (65.3) [a] | |
| > 80 yrs. | 739 (8.7) | 320 (14.9) [a] | 12 (6.1) [b] | |
| Female sex | 2,562 (30.1) | 617 (28.7) | 67 (34.2) | 0.175 |
| Traffic accident | 2,231 (26.2) | 429 (19.9) [a] | 22 (11.2) [a,b] | <0.001 |
| Hospital level | | | | 0.001 |
| Medical center | 2,072 (24.4) | 506 (23.5) | 56 (28.6) | |
| Region hospital | 3,940 (46.3) | 1,055 (49.0) | 106 (54.1) | |
| District hospital or clinics | 2,490 (29.3) | 591 (27.5) | 34 (17.3) [a,b] | |
| Hospital volume (surgeries) | | | | 0.024 |
| Q1 (1–130) | 2,190 (25.8) | 529 (24.6) | 31 (15.8) [a,b] | |
| Q2 (132–262) | 2,067 (24.3) | 517 (24.0) | 60 (30.6) | |
| Q3 (281–544) | 2,175 (25.6) | 585 (27.2) | 49 (25.0) | |
| Q4 (598–1427) | 2,070 (24.3) | 521 (24.2) | 56 (28.6) | |
| Monthly income, USD | | | | 0.004 |
| ≤ $596 | 2,687 (31.6) | 719 (33.4) | 63 (32.1) | |
| $597–$760 | 3,424 (40.3) | 905 (42.1) | 68 (34.7) | |
| > $760 | 2,391 (28.1) | 528 (24.5) [a] | 65 (33.2) [b] | |
| Urbanization level | | | | 0.121 |
| Low | 1,147 (13.5) | 302 (14.0) | 23 (11.7) | |
| Moderate | 2,887 (34.0) | 771 (35.8) | 82 (41.8) | |
| High | 2,458 (28.9) | 607 (28.2) | 46 (23.5) | |
| Very high | 2,010 (23.6) | 472 (21.9) | 45 (23.0) | |
| Comorbidity | | | | |
| Stroke | 936 (11.0) | 448 (20.8) [a] | 54 (27.6) [a] | <0.001 |
| COPD | 484 (5.7) | 198 (9.2) [a] | 15 (7.7) | <0.001 |
| Heart failure | 987 (11.6) | 465 (21.6) [a] | 88 (44.9) [a,b] | <0.001 |
| Coronary heart disease | 1,135 (13.4) | 491 (22.8) [a] | 71 (36.2) [a,b] | <0.001 |
| Hyperlipidemia | 2,166 (25.5) | 748 (34.8) [a] | 54 (27.6) | <0.001 |
| Cardiac dysrhythmia | 373 (4.4) | 156 (7.2) [a] | 11 (5.6) | <0.001 |
| Myocardial infarction | 374 (4.4) | 152 (7.1) [a] | 27 (13.8) [a,b] | <0.001 |
| Hypertension | 4,774 (56.2) | 1,623 (75.4) [a] | 161 (82.1) | <0.001 |
| Dementia | 292 (3.4) | 127 (5.9) [a] | 2 (1.0) [b] | <0.001 |
| Osteoporosis | 938 (11.0) | 299 (13.9) [a] | 19 (9.7) | 0.001 |
| CCI score | 2.0 ± 2.0 | 3.0 ± 2.0 [a] | 6.0 ± 2.0 [a,b] | <0.001 |
| Follow up duration (years) | 6.0 ± 4.2 | 4.1 ± 3.4 [a] | 2.7 ± 2.6 [a,b] | <0.001 |

CCI = Charlson comorbidity index; CKD = chronic kidney disease; COPD = chronic obstructive pulmonary disease; Q = quartile; STD = standardized difference; USD = United States Dollar.

Data were expressed as mean ± standard deviation or frequency (percentage).

"a" and "b" indicate significantly different from "non-CKD" and "non-dialysis CKD" groups, respectively, in the Bonferroni multiple comparisons.

## Early in-hospital outcomes

At index admission, the non-dialysis CKD group had a higher risk of UTI, blood transfusion, ICU stays, in-hospital death; also longer hospital days, and higher medical costs after PHF

**Table 2. In-hospital outcomes of diabetic patients with proximal humerus fracture according to renal function status.**

| Outcome | Number of events (%) | | | Adjusted OR or β ‡ | | | | | |
|---|---|---|---|---|---|---|---|---|---|
| | Non-CKD (n = 8,502) | Non-dialysis CKD (n = 2,152) | Dialysis (n = 196) | Non-dialysis CKD vs. Non-CKD | | Dialysis vs. Non-CKD | | Dialysis vs. Non-dialysis CKD | |
| | | | | OR or β (95% CI) | P | OR or β (95% CI) | P | OR or β (95% CI) | P |
| Categorical | | | | | | | | | |
| Newly-onset VTE | 3 (0.04) | 0 (0.0) | 0 (0.0) | NA | NA | NA | NA | NA | NA |
| Delirium | 5 (0.06) | 3 (0.14) | 0 (0.0) | 1.43 (0.32–6.45) | 0.639 | NA | NA | NA | NA |
| Debridement | 59 (0.69) | 18 (0.84) | 1 (0.51) | 1.50 (0.86–2.61) | 0.151 | 0.81 (0.11–6.05) | 0.833 | 0.54 (0.07–4.15) | 0.550 |
| Infection | 12 (0.14) | 0 (0.0) | 0 (0.0) | NA | NA | NA | NA | NA | NA |
| UTI | 125 (1.5) | 58 (2.7) | 2 (1.0) | 1.59 (1.15–2.21) | 0.006 | 0.79 (0.19–3.28) | 0.746 | 0.50 (0.12–2.08) | 0.339 |
| Pneumonia | 48 (0.57) | 25 (1.2) | 1 (0.51) | 1.64 (0.98–2.74) | 0.061 | 0.84 (0.11–6.35) | 0.864 | 0.51 (0.07–3.91) | 0.518 |
| Transfusion | 1,929 (22.7) | 808 (37.5) | 88 (44.9) | 1.82 (1.63–2.04) | <0.001 | 2.77 (2.02–3.80) | <0.001 | 1.52 (1.10–2.10) | 0.011 |
| ICU stay | 226 (2.7) | 86 (4.0) | 15 (7.7) | 1.36 (1.04–1.77) | 0.027 | 2.20 (1.22–3.95) | 0.008 | 1.62 (0.89–2.97) | 0.116 |
| In-hospital death | 19 (0.22) | 15 (0.70) | 1 (0.51) | 2.34 (1.15–4.77) | 0.020 | 2.07 (0.25–16.81) | 0.497 | 0.89 (0.11–7.16) | 0.909 |
| Continuous | | | | | | | | | |
| Hospital days | 7.0 ± 13.0 | 8.0 ± 12.0 | 9.0 ± 8.0 | 0.77 (0.14, 1.41) | 0.016 | 1.63 (-0.23, 3.50) | 0.086 | 0.86 (-1.05, 2.77) | 0.378 |
| Cost (USD×10³) | 4.9 ± 6.4 | 5.6 ± 8.4 | 7.6 ± 8.1 | 0.20 (0.09, 0.31) | <0.001 | 0.83 (0.51, 1.15) | <0.001 | 0.63 (0.30, 0.96) | <0.001 |

β = regression coefficient; CI = confidence interval; CKD = chronic kidney disease; ICU = intensive care unit; NA = not applicable; OR = odds ratio; USD = United States Dollar; UTI = urinary tract infection; VTE = venous thromboembolism.

Data were expressed as frequency (percentage) or mean ± standard deviation.

‡The model was adjusted for all covariates listed in Table 1, in which the follow-up year was replaced with the index date.

fixation surgery when compared with the non-CKD group. Similarly, the dialysis group also had a higher risk of blood transfusion and ICU stay and higher medical cost than the non-CKD group. Also, the dialysis group had a higher risk of blood transfusion and higher medical cost when compared to the non-dialysis CKD group (Table 2).

## Late perioperative outcomes

Table 3 shows the results of late outcomes. The non-dialysis CKD group had a higher risk of any infection after PHF fixation surgery compared to the non-CKD group (sub-distribution hazard ratio [SHR], 1.51; 95% confidence interval [CI]: 1.18–1.93). The dialysis group also had a higher risk of any infection compared to the non-dialysis CKD group (SHR, 1.90; 95% CI: 1.14–3.19) and the non-CKD group (SHR, 2.86; 95% CI: 1.73–4.74) (Fig 2A). The result of all-cause revision was similar to that of overall infection (Fig 2B).

At any stage (30 days, 90 days, one year, or at the end of follow-up), the non-dialysis CKD group had a higher risk of all-cause readmission compared to the non-CKD group; the dialysis group also had a higher risk of all-cause readmission compared to the non-dialysis CKD group (Fig 2C). In the early stage (at 30 days), the non-dialysis CKD group had a higher risk of all-cause mortality than the non-CKD group. After that (including at 90 days, one-year, and at the

**Table 3. Late outcomes of diabetic patients with proximal humerus fracture according to renal function status.**

| Outcome | Number of events (%) | | | Adjusted model‡ | | | | | |
|---|---|---|---|---|---|---|---|---|---|
| | Non-CKD (*n* = 8,502) | Non-dialysis CKD (*n* = 2,152) | Dialysis (*n* = 196) | Non-dialysis CKD *vs.* Non-CKD | | Dialysis *vs.* Non-CKD | | Dialysis *vs.* Non-dialysis CKD | |
| | | | | HR or SHR (95% CI) | *P* | HR or SHR (95% CI) | *P* | HR or SHR (95% CI) | *P* |
| Infection event | | | | | | | | | |
| Superficial infection | 264 (3.1) | 85 (4.0) | 15 (7.7) | 1.48 (1.14–1.93) | 0.004 | 2.71 (1.57–4.67) | <0.001 | 1.83 (1.04–3.20) | 0.035 |
| Debridement | 49 (0.58) | 14 (0.65) | 1 (0.51) | 1.41 (0.74–2.67) | 0.300 | 1.03 (0.13–7.99) | 0.975 | 0.74 (0.09–5.74) | 0.770 |
| Removal | 14 (0.17) | 10 (0.47) | 3 (1.53) | 3.79 (1.46–9.86) | 0.006 | 8.27 (1.89–36.08) | 0.005 | 2.18 (0.60–7.95) | 0.238 |
| Overall infection | 299 (3.5) | 97 (4.5) | 18 (9.2) | 1.51 (1.18–1.93) | 0.001 | 2.86 (1.73–4.74) | <0.001 | 1.90 (1.14–3.19) | 0.015 |
| Revision | | | | | | | | | |
| Infectious revision | 6 (0.07) | 6 (0.28) | 1 (0.51) | 4.44 (1.34–14.7) | 0.015 | 5.10 (0.48–54.72) | 0.178 | 1.15 (0.11–12.25) | 0.908 |
| Non-infectious revision | 220 (2.6) | 72 (3.4) | 13 (6.6) | 1.52 (1.16–2.00) | 0.003 | 3.49 (1.93–6.32) | <0.001 | 2.30 (1.25–4.22) | 0.007 |
| All-cause revision | 226 (2.66) | 78 (3.6) | 14 (7.1) | 1.60 (1.23–2.09) | 0.001 | 3.57 (2.02–6.33) | <0.001 | 2.23 (1.25–4.01) | 0.007 |
| All-cause readmission | | | | | | | | | |
| At 30 days | 675 (8.0) | 273 (12.8) | 45 (23.1) | 1.42 (1.23–1.65) | <0.001 | 2.77 (2.00–3.82) | <0.001 | 1.95 (1.40–2.71) | <0.001 |
| At 90 days | 1,464 (17.3) | 581 (27.2) | 87 (44.6) | 1.50 (1.35–1.66) | <0.001 | 2.68 (2.12–3.40) | <0.001 | 1.79 (1.41–2.28) | <0.001 |
| At 1 year | 3,642 (42.9) | 1,099 (51.4) | 132 (67.7) | 1.28 (1.19–1.38) | <0.001 | 1.94 (1.59–2.36) | <0.001 | 1.51 (1.24–1.85) | <0.001 |
| At the end | 6,469 (76.3) | 1,680 (78.6) | 168 (86.2) | 1.26 (1.19–1.33) | <0.001 | 1.72 (1.44–2.07) | <0.001 | 1.37 (1.14–1.65) | 0.001 |
| All-cause mortality | | | | | | | | | |
| At 30 days | 52 (0.61) | 35 (1.6) | 2 (1.0) | 1.93 (1.23–3.01) | 0.004 | 1.80 (0.42–7.62) | 0.426 | 0.93 (0.22–3.97) | 0.925 |
| At 90 days | 104 (1.2) | 72 (3.3) | 11 (5.6) | 1.96 (1.43–2.69) | <0.001 | 4.21 (2.18–8.10) | <0.001 | 2.14 (1.11–4.13) | 0.023 |
| At 1 year | 306 (3.6) | 192 (8.9) | 37 (18.9) | 1.93 (1.60–2.33) | <0.001 | 5.29 (3.68–7.60) | <0.001 | 2.74 (1.90–3.95) | <0.001 |
| At the end | 2,086 (24.5) | 761 (35.4) | 110 (56.1) | 1.58 (1.44–1.72) | <0.001 | 4.31 (3.53–5.27) | <0.001 | 2.74 (2.23–3.36) | <0.001 |

CI = confidence interval; CKD = chronic kidney disease; HR = hazard ratio; SHR = sub-distribution hazard ratio.

Data were given as frequency (percentage).

‡The model was adjusted for all covariates listed in Table 1, in which the follow-up year was replaced with the index date.

end of follow-up), the dialysis group showed a higher risk of all-cause mortality when compared to the other two groups (Fig 2D).

**Subgroup analysis by baseline characteristics.** The association between renal dysfunction and the risk of overall infection was more evident in those with traffic accidents (*P* for interaction = 0.086) and lower CCI scores (*P* for interaction = 0.096) (Fig 3A). None of the baseline characteristics significantly modified the association between renal dysfunction and the risk of all-cause revision (Fig 3B) and all-cause readmission (Fig 3C). However, the association between renal dysfunction and the risk of all-cause mortality were modified by females (*P* for interaction = 0.011), those living at higher urbanization level (*P* for interaction = 0.057), and those with higher CCI scores (*P* for interaction = 0.069) (Fig 3D).

## Discussion

This population-based cohort study analyzed the outcomes following proximal humerus fracture fixation surgery in diabetic patients with different renal statuses. Compared to those without dialysis, dialysis patients bore higher risks of infection, revision, readmission, and mortality. Similarly, the non-dialysis CKD group had a higher risk of infection, revision,

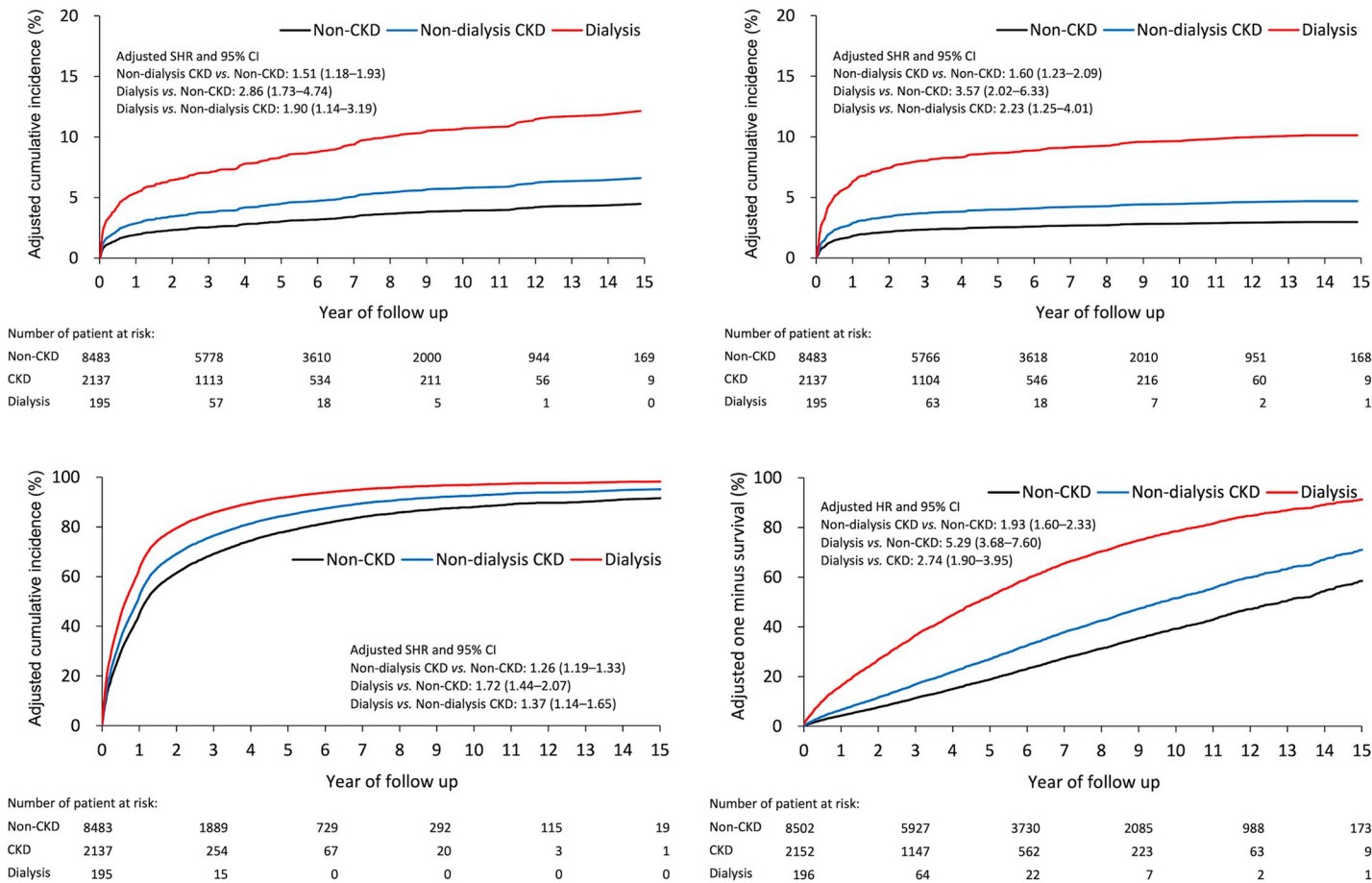

**Fig 2. The direct-adjusted cumulative incidence of outcomes among T2DM patients admitted due to proximal humerus fracture, by renal function status.** (A) Overall infection. (B) All-cause revision. (C) All-cause readmission. (D) The direct-adjusted survival rate of all-cause mortality. CKD = chronic kidney disease; SHR = sub-distribution hazard ratio; CI = confidence interval; HR = hazard ratio.

readmission, and mortality at all times compared to the non-CKD group. The deteriorated renal function was associated with elevated risks for infection, revision, readmission, and mortality after proximal humerus fracture fixation surgery.

Additional subgroup analysis showed that some baseline characteristics might modify CKD's effect on the outcomes, including infection and mortality. Diabetic patients with CKD had a higher incidence of postoperative infection than those without CKD in patients with traffic accidents or those with a lower CCI score ($< 3$). Furthermore, diabetic patients with CKD had a higher incidence of mortality than those without CKD in females, residents in higher urbanization areas, or higher CCI scores ($\geq 3$).

Dialysis has been identified as a risk factor for postoperative complications following major surgeries [29]. CKD patients have a greater risk of surgical site infection and wound complications than those with normal renal function. Furthermore, surgical site infection risk is even higher in CKD patients receiving dialysis [30]. In the current study, dialysis patients had a higher infection rate and poorer wound healing than the other two groups. This postoperative infection could be attributed to immuno-compromise, renal anemia, nutritional deficiencies, and poor circulation. The subgroup analysis showed that CKD's effect on infection was more significant in patients with fewer comorbidities (CCI<3) and those with traffic accidents. The

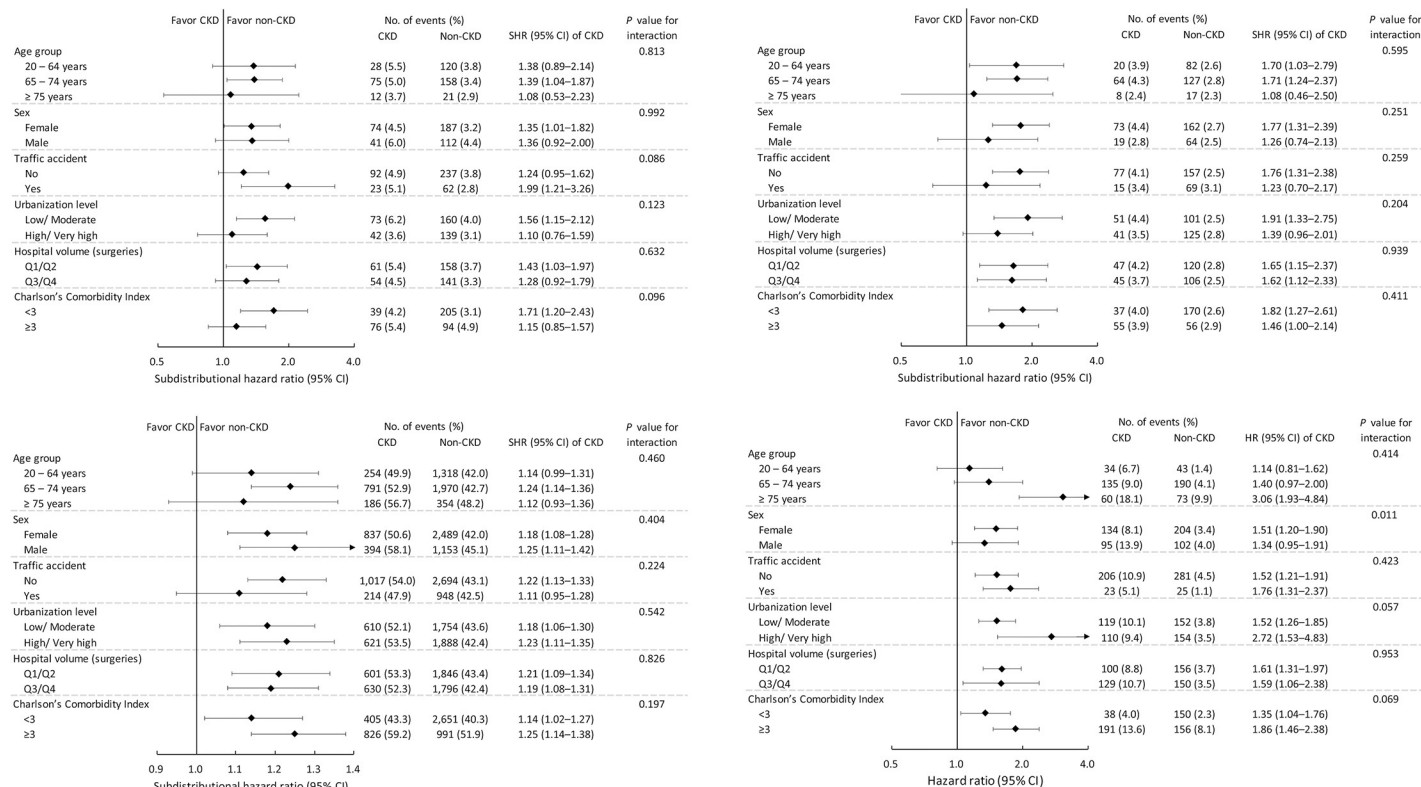

**Fig 3. Subgroup analysis, comparing the risk of outcomes between non-CKD and CKD patients.** (A) Overall infection (B) All-cause revision. (C) All-cause readmission. (D) All-cause mortality. Due to the small sample size of the dialysis group, the non-dialysis CKD group and dialysis group have been combined as the CKD group in the subgroup analysis.

Charlson comorbidity index encompasses multiple factors with a propensity for infection following fractures, such as diabetes, hepatitis C, acquired immunodeficiency syndrome, and cancer with immunomodulation therapy [31]. Although CKD also increases the risk of infection, CKD's marginal effect was less predominant in patients with multiple comorbidities. Besides, the delayed surgery was a risk factor for infection after a fracture [32]. Fractures from traffic accidents might occur with contaminant injuries such as the head, chest, or abdominal trauma, which further delayed surgical osteosynthesis.

Readmission was an issue for CKD patients, jeopardizing the quality of life and increasing the healthcare burden. In the international Dialysis Outcomes and Practice Patterns Study (DOPPS), dialysis was associated with higher fracture and admission rates than in the general population (9). Advanced age, poor compliance, and comorbidities have been validated as predictors of admission for causes other than vascular access related in dialysis patients [33]. In the current study, although being younger, the dialysis group still had a significantly higher readmission rate than the non-CKD group. Therefore, the comorbidities matter more than the age in the event of readmission.

The one-year mortality in the current study was about 5%, similar to that previously reported for the general population's proximal humerus fracture [15, 34]. Clement et al. reported a 10% one-year mortality in the UK population [34]. Hence, it could be concluded that caring quality in Taiwan might be comparable to other countries. In this study, 8.9% of CKD patients and 18.9% of dialysis patients died one year after surgery. Thus, CKD was an independent risk factor for elevated one-year mortality following proximal humerus fracture,

no matter whether there was dialysis or not. Similar to our study, two nationwide database studies showed hemodialysis patients had a 3.7 to 4.8-fold mortality risk increase after major fractures than the general population [9, 35]. Therefore, CKD patients should be given more attentive care, given the increased mortality risk after a fracture episode.

The fracture location may matter. Compared to our previous studies [10, 11], the one-year mortality rate following proximal humerus fracture might be lower than that following hip fracture. This finding was slightly different from that of another previous Asian study [35]. Mandai et al. [35] demonstrated that upper arm fracture was associated with 2-times higher odds of vascular access failure in dialysis patients. Therefore, dialysis patients with upper arm fractures had a slightly higher 30-day mortality risk compared to those with hip fractures. However, we did not directly compare the outcomes following different fractures. Further study comparing fractures of different anatomical locations is needed to elucidate this issue further.

The subgroup analysis shows that the mortality rate is significantly associated with CKD in female patients, those living in more urbanized areas, and more comorbidities (CCI≥3). It is still unclear whether fracture mortality varies with the degree of urbanization. Diamantopoulos et al. found no significant differences in either 1-year or 5-year mortality between rural and urban areas [36]. Weller, et al. found a trend toward increased in-hospital and one-year mortality in rural areas, compared to urban community hospitals [37]. This difference resulted from a higher number of specialized personnel and the high volume of patients treated in urban hospitals.

In contrast, a Norwegian nationwide study found an increasing degree of urbanization was associated with higher post-fracture mortality, especially the first 1 to 2 years post-fracture [38], which might reflect a higher level of fragility in urban residents. Another Norwegian study demonstrated that urban women had lower BMD, lower BMI, and a higher proportion of poor or fair self-reported health than rural women [39]. The above findings were consistent with our study, whereby CKD poses a risk of mortality for residents in more urbanized regions. The US National Hospital Discharge Survey (NHDS) and one cohort found comorbidities and malignancy significant risk factors of mortality following a proximal humeral fracture [40]. Our study further supports that multiple comorbidity is associated with mortality following proximal humerus fracture.

Our study's strengths include the nationwide coverage of the NHIRD database, consisting of nearly all patients in Taiwan. In addition, we included patients across institutions and living regions, regardless of admission to hospitals or nursing facilities. This fact increases the potential clinical applicability of the study. However, our study had some limitations too. The NHIRD database did not contain laboratory data or medical images, preventing us from further differentiating renal function and fracture patterns. The reduction quality, implant choice, and position were not mentioned, either. However, the strict regulations of NHIRD for dialysis-dependent patients ensured reliable data.

## Conclusion

Deteriorated renal function is associated with elevated risks for infection, revision, readmission, and mortality after proximal humerus fracture surgery in diabetic patients. Therefore, these findings should be taken into consideration when caring for diabetic patients.

## Supporting information

**S1 Table. ICD-9-CM code used for diagnosis in the current study.**
(DOCX)

## Acknowledgments

We thank Mr. Alfred Hsing-Fen Lin and Mrs. Bing-Yu Chen Ph.D. at Raising Statistic Consultant Inc., for their assistance with the statistical analysis during the completion of the manuscript.

## Author Contributions

**Conceptualization:** Chien-Tien Chen, Su-Ju Lin, Liang-Tseng Kuo.

**Data curation:** Chi-Lung Chen.

**Formal analysis:** Liang-Tseng Kuo.

**Investigation:** Pei-An Yu.

**Methodology:** Su-Ju Lin, Tien-Hsing Chen, Chi-Lung Chen.

**Project administration:** Kuo-Ti Peng.

**Resources:** Tien-Hsing Chen.

**Supervision:** Wei-Hsiu Hsu.

**Validation:** Yao-Hung Tsai.

**Visualization:** Pei-An Yu.

**Writing – original draft:** Chien-Tien Chen.

**Writing – review & editing:** Liang-Tseng Kuo, Wei-Hsiu Hsu, Kuo-Ti Peng.

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
