## [Decision Letter · Decision Letter 0]

19 Aug 2021

PONE-D-21-21781

Effect of Chronic Kidney Disease on Outcomes Following Proximal Humerus Fragility Fracture Surgery in Diabetic Patients: A Nationwide Population-based Cohort Study

PLOS ONE

Dear Dr. Kuo,

Thank you for submitting your manuscript to PLOS ONE. After careful consideration, we feel that it has merit but does not fully meet PLOS ONE’s publication criteria as it currently stands. Therefore, we invite you to submit a revised version of the manuscript that addresses the points raised during the review process.

Reviewer 2 requests a few clarifications of your text.  Since the journal does not copy edit, this is your responsibility.

We look forward to receiving your revised manuscript.

Kind regards,

Robert Daniel Blank, MD, PhD

Academic Editor

PLOS ONE

Journal Requirements:

"NO"

Reviewers' comments:

Reviewer's Responses to Questions

**Comments to the Author**

1. Is the manuscript technically sound, and do the data support the conclusions?

Reviewer #1: Yes

Reviewer #2: Yes

2. Has the statistical analysis been performed appropriately and rigorously? 

Reviewer #1: Yes

Reviewer #2: Yes

3. Have the authors made all data underlying the findings in their manuscript fully available?

Reviewer #1: Yes

Reviewer #2: Yes

4. Is the manuscript presented in an intelligible fashion and written in standard English?

Reviewer #1: Yes

Reviewer #2: Yes

5. Review Comments to the Author

Reviewer #1: Good review of cases of Proximal Humerus Fracture correlated to CRF and DM.

Possible to change the recommendation of treatment for this group of patient with high risk, from surgical to non-surgical treatment.

Reviewer #2: This is an interesting stud. A few comments for revision:

1. In Figures 2A-D: replace "normal" with non-CKD

2. Line 272 – what population is this referring to? Within dialysis-requiring CKD?

3. Line 276 – What does “this cohort” refer to – the whole registry, or your cohort of diabetes subjects? (given that, in line 279 you then list another set of % of mortality “in our cohort”)

4. Line 290 – higher or lower mortality risk in upper arm fractures?

5. Line 320 – suggest adding in “in diabetes patients” in first sentence

6. PLOS authors have the option to publish the peer review history of their article (what does this mean?). If published, this will include your full peer review and any attached files.

Reviewer #1: No

Reviewer #2: No

---

## [Author Response · Author response to Decision Letter 0]

4 Sep 2021

Please see "response to reviewer" file.

---

## [Decision Letter · Decision Letter 1]

27 Sep 2021

Effect of Chronic Kidney Disease on Outcomes Following Proximal Humerus Fragility Fracture Surgery in Diabetic Patients: A Nationwide Population-based Cohort Study

PONE-D-21-21781R1

Dear Dr. Kuo,

We’re pleased to inform you that your manuscript has been judged scientifically suitable for publication and will be formally accepted for publication once it meets all outstanding technical requirements.

Kind regards,

Robert Daniel Blank, MD, PhD

Academic Editor

PLOS ONE

Additional Editor Comments (optional):

Reviewers' comments:

Reviewer's Responses to Questions

**Comments to the Author**

1. If the authors have adequately addressed your comments raised in a previous round of review and you feel that this manuscript is now acceptable for publication, you may indicate that here to bypass the “Comments to the Author” section, enter your conflict of interest statement in the “Confidential to Editor” section, and submit your "Accept" recommendation.

Reviewer #2: (No Response)

2. Is the manuscript technically sound, and do the data support the conclusions?

Reviewer #2: (No Response)

3. Has the statistical analysis been performed appropriately and rigorously? 

Reviewer #2: (No Response)

4. Have the authors made all data underlying the findings in their manuscript fully available?

Reviewer #2: (No Response)

5. Is the manuscript presented in an intelligible fashion and written in standard English?

Reviewer #2: (No Response)

6. Review Comments to the Author

Reviewer #2: (No Response)

7. PLOS authors have the option to publish the peer review history of their article (what does this mean?). If published, this will include your full peer review and any attached files.

Reviewer #2: No

---

## [Editor Report · Acceptance letter]

30 Sep 2021

PONE-D-21-21781R1 

Effect of Chronic Kidney Disease on Outcomes Following Proximal Humerus Fragility Fracture Surgery in Diabetic Patients: A Nationwide Population-based Cohort Study 

Dear Dr. Kuo:

I'm pleased to inform you that your manuscript has been deemed suitable for publication in PLOS ONE. Congratulations! Your manuscript is now with our production department. 

Kind regards, 

on behalf of

Professor Robert Daniel Blank 

Academic Editor

PLOS ONE